# A novel *canis lupus familiaris* reference genome improves variant resolution for use in breed-specific GWAS

Robert A Player[1], Ellen R Forsyth[1], Kathleen J Verratti[1], David W Mohr[2], Alan F Scott[2], Christopher E Bradburne[1,2]

**Reference genome fidelity is critically important for genome wide association studies, yet most vary widely from the study population. A typical whole genome sequencing approach implies short-read technologies resulting in fragmented assemblies with regions of ambiguity. Further information is lost by economic necessity when genotyping populations, as lower resolution technologies such as genotyping arrays are commonly used. Here, we present a phased reference genome for *Canis lupus familiaris* using high molecular weight DNA-sequencing technologies. We tested wet laboratory and bioinformatic approaches to demonstrate a minimum workflow to generate the 2.4 gigabase genome for a Labrador Retriever. The de novo assembly required eight Oxford Nanopore R9.4 flowcells (~23X depth) and running a 10X Genomics library on the equivalent of one lane of an Illumina NovaSeq S1 flowcell (~88X depth), bringing the cost of generating a nearly complete reference genome to less than $10K (USD). Mapping of short-read data from 10 Labrador Retrievers against this reference resulted in 1% more aligned reads versus the current reference (Can-Fam3.1, *P* < 0.001), and a 15% reduction of variant calls, increasing the chance of identifying true, low-effect size variants in a genome-wide association studies. We believe that by incorporating the cost to produce a full genome assembly into any large-scale genotyping project, an investigator can improve study power, decrease costs, and optimize the overall scientific value of their study.**

## Introduction

The revolution in genomic sequencing technologies is creating a wealth of information about diverse taxa. Typically, an organism is sequenced as a high quality reference, and then the variability in genomic content within individuals is surveyed using cheaper, more economically viable technologies (Green & Guyer, 2011). Over time, the costs of genomic characterization are reduced as technological

performance increases. This means that periodically, new references need to be established that can be used for read mapping and scaled genotyping approaches, such as the design of new single-nucleotide polymorphism (SNP) arrays used to genotype large numbers of individuals. An example is the human genome, which was established in the draft form in 2001 at a cost of $3.2B US (Venter et al, 2001). After completion, haplotyping of populations continued at a large scale using high-throughput SNP chips, which initially started with a few hundred 1,000 SNPs but within 10 yr contained millions. Likewise the human reference has been continually updated. Starting in 2001, the draft sequence covered more than 90% of the genome, had a 1:1,000 bp error rate, and contained 150,000 gaps. Within 2 yr, the same genome had reached 99% coverage, 1:10,000 bp error rate, and only 400 gaps ("Human Genome Project FAQ," n.d.). According to the National Human Genome Research Institute tracking site, the cost has stabilized at around $1,000 per full human genome since 2015. However, the human genomes considered for this estimation do not come close to full completion, having a 1:100 bp error rate along with widely varying percent coverage ("DNA Sequencing Costs: Data," n.d.). The $1,000 estimate also assumes the utilization of whole-genome sequencing (WGS) short-read technologies. For genome-wide association studies (GWAS), lack of genetic information due to incomplete genomes can lead to false negatives from an inability to see real variants or false positives from false variant calls against a reference. In fact, the early reliance on SNPs to type the variation in humans has likely contributed to the "missing heritability" problem of human genomic medicine (Manolio et al, 2009; Young, 2019).

Canids share a similar story. The current reference sequence for canids is a boxer: CanFam3.1, submitted to the National Center for Biotechnology Information (NCBI) in November 2011 (Kim et al, 1998; Lindblad-Toh et al, 2005). It was sequenced mostly based on Sanger shotgun sequencing with limited Illumina polishing, and the annotations have been continuously updated (the latest update as of this article was in June 2019) ("Canis Lupus Familiaris - Ensembl Genome Browser 100," n.d.). Various SNP genotyping chips, whose costs are dependent on scale but average $100–$500 per animal, have been developed, but much of the detectable genetic

---

[1]Asymmetric Operations Sector, The Johns Hopkins University Applied Physics Laboratory, Laurel, MD, USA    [2]McKusick-Nathans Department of Genetic Medicine, Johns Hopkins School of Medicine, Baltimore, MD, USA

Correspondence: chris.bradburne@jhuapl.edu

variation depends on an incomplete and constantly changing reference. Long read technologies have the potential to change this paradigm and lead the community to generate single reference genomes for individual projects. The longer read lengths of ~2–30 kb remove many of the bioinformatic challenges inherent in short read sequencing and allow previously unheard of resolution to observe structural variants and the organization of long stretches of low-complexity DNA. A genome assayed with this "high-resolution genomic" approach using longer reads could provide structural variants together with SNPs. Furthermore, application of high-resolution genomics across a population for a GWAS could illuminate any "missing heritability" for a population, such as structural variants that are unresolvable with SNP or WGS short-read platforms. Canids provide an excellent test case for this approach.

*Canis lupus familiaris* has been under selection by human breeding for thousands of years, which has created extremely variable morphologies within a single species (Plassais et al, 2019). Therefore, unlike human genomes that have many common variants of low effect size, dogs have many common variants of large effect size. Any study that lacks genomic context of a breed by not having a high-quality reference genome specific to that breed runs the risk of missing important SNPs and structural variants that may be associated with interesting phenotypes. We set out to establish the best laboratory and bioinformatic workflows to provide the highest quality genome at the lowest cost, taking advantage of Oxford Nanopore Technologies (ONT), 10X Genomics, and Illumina sequencing technologies. The resulting genome is of a male yellow Labrador Retriever, named "Yella," and we estimate that similar workflows could be used to easily generate high-quality reference genomes for researchers or breeders establishing studies requiring high-resolution variation. Furthermore, we assert that any large-scale study on genetic variation for a population should begin with the establishment of a high-quality reference genome for that population.

# Results

When setting out to produce a high-quality, phased reference genome, careful consideration should be given to wet laboratory processes that do the following: (1) provide optimal preservation for downstream extraction, (2) generate high quantity and quality of high molecular weight (HMW) DNA, and (3) are robust and reproducible (i.e., they provide the least amount of variability between different individual blood samples). Fig 1 shows the wet laboratory process flow and components that were evaluated in this study, and used to generate HMW canine DNA for sequencing and de novo genome assembly.

### Preservation, extraction, and acquisition of HMW-DNA

Canine blood samples were collected and delivered in either the PAXgene DNA proprietary storage media or a purple top vacutainer tube with EDTA. These two preservative types were evaluated in conjunction with four DNA extraction and isolation methods: (1) a standard phenol chloroform extraction (PCE) method, (2) the Magmax Core NA Purification, (3) the Nanobind CBB Big DNA kit, and (4) the PAXgene Blood DNA Kit. Blood samples from Yella stored in the purple top tubes and extracted with the Nanobind kit yielded the best purity (highest 260/280 ratio) and highest concentrations (Table 1, additional information in Table S1). Compared with PCE from the same storage method, this is equivalent to a 92-fold increase in extraction efficiency. In terms of total recovered NA, the PAXgene extraction from the purple top tube performed best, yielding over 10 $\mu$g DNA. Most importantly, significant fractions of HMW-DNA using the PAXgene extraction kit were not detected (Fig S1). Direct

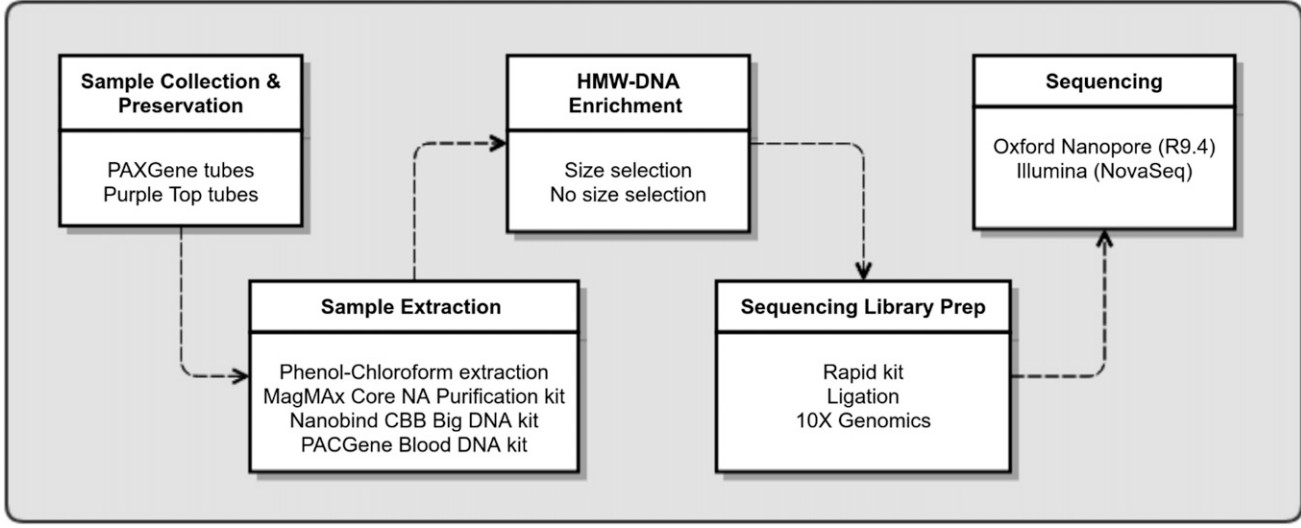

**Figure 1. Diagram of wet laboratory workflow.**
Sample collection, extraction, and sequencing library preparation methods used in this study are shown.

**Table 1.   Effect of blood sample preservation agent on DNA yield.**

| Storage agent | NA isolation method | Input volume (uL) | Output volume (uL) | NA conc (ng/uL) | Recovered NA total (ng) | NA total per mL blood | NA quality (260/280) | HMW DNA yielded? |
|---|---|---|---|---|---|---|---|---|
| Proprietary (PAXgene) | PCE | 1,700 | 1,000 | 6.37 | 6,370 | 3,747 | 2.20 | Yes |
| Proprietary (PAXgene) | Magmax Core NA purification | 200 | 90 | 2.03 | 183 | 914 | 1.66 | Yes |
| Proprietary (PAXgene) | Nanobind CBB Big DNA kit | 200 | 100 | 11.10 | 1,110 | 5,550 | 1.87 | Yes |
| Proprietary (PAXgene) | PAXgene Blood DNA kit | 1,700 | 1,000 | 6.40 | 6,400 | 3,765 | 2.38 | No |
| EDTA (purple top) | PCE | 1,700 | 1,000 | 0.38 | 380 | 224 | 5.21 | Yes |
| EDTA (purple top) | Magmax Core NA Purification | 200 | 90 | 2.63 | 237 | 1,184 | 1.62 | Yes |
| EDTA (purple top) | Nanobind CBB Big DNA kit | 200 | 100 | 35.30 | 3,530 | 17,650 | 1.84 | Yes |
| EDTA (purple top) | PAXgene Blood DNA kit | 1,700 | 1,000 | 10.80 | 10,800 | 6,353 | 1.98 | No |

Blood for one canine (Yella) was drawn directly into two tubes containing either a proprietary preservation agent, or EDTA. Three kits were tested against a phenol-chloroform extraction (PCE) standard method. Input and output volumes for each kit are shown, along with actual recovered total DNA mass. NA stands for nucleic acid. EDTA stands for ethylenediaminetetraacetic acid. The extremely high quality (5.21) observed for PCE is likely due to the presence of residual phenol in some samples, which is known to increase the 260/280 ratio beyond the normal quality range.

comparison of extraction kits showed that the Nanobind kit provided the most consistent DNA yield (Fig 2A) and quality (Fig 2B) among the four kits tested using blood stored in EDTA from four different canines (Table 2).

### DNA size-selection and Oxford nanopore sequencing

Estimated average genome depth, based on the 2.32 Gb (gigabase) CanFam3.1 genome, for combined read data from all eight ONT R9.4.1 flowcells was 22.65× (Table 3). Additional read statistics for the combined read data are shown in Fig S2. The read N50 varied per flow cell dataset from 11,868 to 35,584 bp (Table 4). Interestingly, size selection with the Circulomics Short Read Eliminator kit before library preparation did not always result in a higher read N50, and in fact the read N50 was actually reduced when the kit was used before library preparation with the ligation kit (SQK-LSK109). Instead, read N50 appears more influenced by library kit type, with the ligation kit having ~2× higher median read N50 than the rapid kit

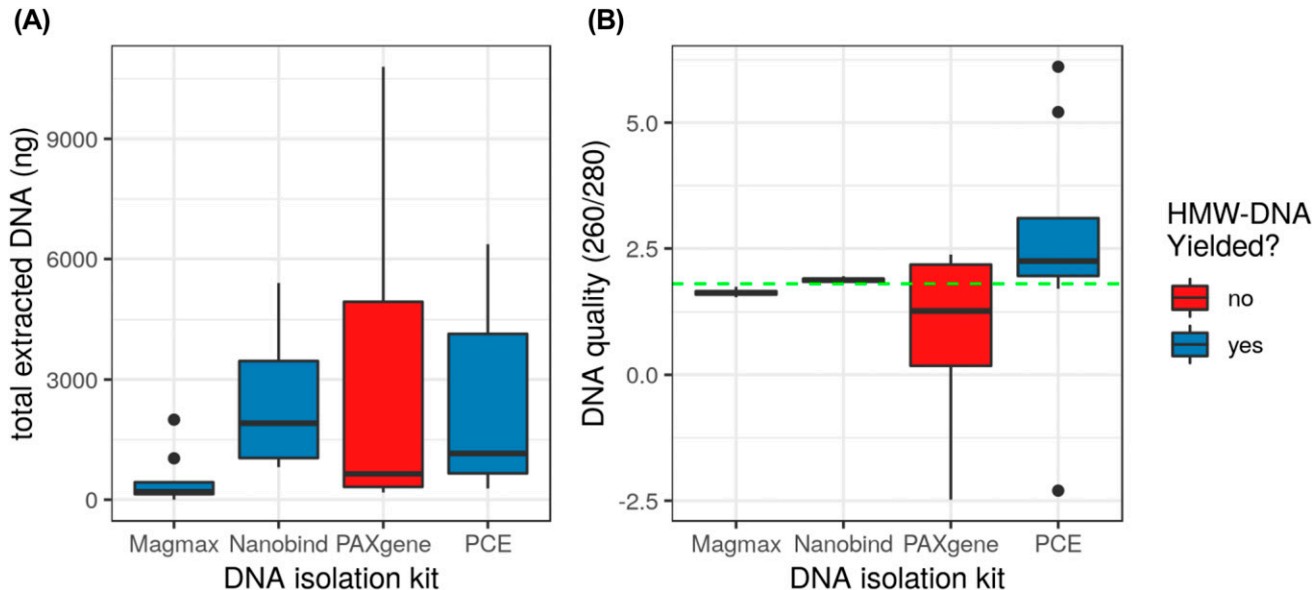

**Figure 2.   Total extracted DNA and DNA quality from four tested isolation kits.**
**(A)** Total extracted DNA. **(B)** DNA quality; green line indicates the ideal 260/280 ratio for DNA purity at 1.80. Extractions from the Nanobind kit had the most consistently high yield and quality.

**Table 2. Variability of NA (nucleic acid) isolation method across four canine blood samples preserved in "purple top" tubes with EDTA.**

| NA isolation method | Total NA (ug) mean | Total NA Std. Dev. | NA quality (260/280) mean | NA quality (260/280) Std. Dev. | High-MW DNA? |
|---|---|---|---|---|---|
| PCE | 1.28 | 1.63 | 1.75 | 3.10 | **Yes** |
| Magmax Core NA purification | 0.81 | **1.02** | 1.57 | 0.05 | **Yes** |
| Nanobind CBB Big DNA kit | 2.92 | 1.99 | **1.85** | **0.03** | Yes |
| PAXgene Blood DNA kit | **4.08** | 4.85 | 1.73 | 0.79 | No |

DNA from purple top tubes was extracted using either phenol–chloroform extraction (PCE), or three commercial kits (Magmax, Nanobind, and PAXgene). Bold values represent the best performance in a particular category.

(SQK-RAD004) (median read N50 of 24,750 and 12,094 bp, respectively).

### Illumina sequencing of 10X Genomics library and SuperNova scaffolding

The estimated average genome depth for trimmed 2 × 150 bp reads from the prepared 10X Genomics library run on an Illumina NovaSeq 6000 S1 flowcell was 87.80× (Table 5). SuperNova scaffolding was performed, which uses the 10X GEM barcoding preparation for more accurate localization of short reads into contigs, under the assumption that reads sharing the same barcode are derived from the same small number of HMW DNA fragments contained in each GEM. The resulting scaffold contained 10,391 contigs, with a contig N50 and L50 of 94 kb and 22 contigs, respectively. The phase block size was greater than 5 Mb (megabase), and the scaffold N50 was 39 Mb. The assembly size of scaffolds greater than or equal to 10 kb was 2.33 Gb, which is in agreement with other canine breed assemblies such as the Boxer (CanFam3.1 assembly at 2.31 Gb) and German Shepherd (GCA_008641245.1 assembly at 2.36 Gb).

### De novo assembly

The effect of estimated average read depth and library preparation kit (SQK-RAD004 or SQK-LSK109, that is, rapid or ligation, respectively) on assembly contig count and total length was examined. The overriding factor for achieving the expected ~2.35 Gb assembly length is read depth, with the combination of reads from all eight flow cells achieving the expected length and about a magnitude reduction in total contigs compared with the CanFam3.1 assembly. ONT kit type had less of an effect on total length and contig count, with the ligation-only assemblies (at 10.02× depth) achieving a higher total length than the rapid-only assemblies (at 12.64× depth), even at ~2.5× lower estimated depth. However, the ligation kit assemblies appear more influenced by miniasm parameter selection compared with the rapid kit assemblies. A combination of kit types at a similar estimated depth (12.05×) seems to be the best of both worlds, with resulting assemblies having approximately the same number of contigs as the rapid-only assemblies (i.e., lower than ligation-only assemblies) at a comparable total length to the ligation-only assemblies.

Next, the effect of parameters available in the de novo assembler called miniasm on the ~23× estimated genome depth assemblies was assessed by examining the assembly cluster at the top left of Fig 3. Fig 4 shows 144 assemblies, which correspond to 144 unique parameter sets tested. It is important to note, however, that because the "m" parameter had no effect on the assembly attributes of interest, there appears to be only 48 points in each plot. The following correlations and description of effect on assembly attributes is with respect to an increasing parameter value (see Fig 3 legend for description of parameters): m, not correlated, no effect; i, negative correlation, slightly less total bps but more contigs; s, negative correlation, significantly less total bps but more contigs; I, positive correlation, moderately more contigs and total bps; e, positive correlation, less contigs and less total bps.

**Table 3. Breakdown of ONT sequencing runs, flow cells, library kit type, and estimated depth shown in Fig 2.**

| Run # | Flowcell # | ONT kit | Total flow cells | Est. Depth |
|---|---|---|---|---|
| 1 | 1, 2 | SQK-LSK109 | 2 | 6.66 |
| 2 | 5, 6 | SQK-LSK109 | 2 | 3.96 |
| 1 + 2 | 1, 2, 5, 6 | SQK-LSK109 | 4 | 10.02 |
| 1 | 3, 4 | SQK-RAD004 | 2 | 5.99 |
| 2 | 7, 8 | SQK-RAD004 | 2 | 6.65 |
| 1 + 2 | 3, 4, 7, 8 | SQK-RAD004 | 4 | 12.64 |
| 1 | 1, 2, 3, 4 | RAD+LSK | 4 | 12.05 |
| 2 | 5, 6, 7, 8 | RAD+LSK | 4 | 10.6 |
| 1 + 2 | 1, 2, 3, 4, 5, 6, 7, 8 | RAD+LSK | 8 | 22.65 |

Flowcell number from Table 2.

**Table 4.** Oxford Nanopore GridION sequencing run summaries using R9.4.1 flowcells.

| Run | Flowcell # | ONT kit | Total bp | Total reads | Read N50 | Mean quality (Phred) |
|-----|-----------|---------|----------|-------------|----------|----------------------|
| 1 | 1 | SQK-LSK109 | 6,274,113,013 | 658,356 | 22,619 | 11.7 |
| 1 | 2 | SQK-LSK109[a] | 7,769,391,385 | 934,471 | 18,562 | 12.2 |
| 1 | 3 | SQK-RAD004 | 6,301,883,845 | 1,026,445 | 11,868 | 11.9 |
| 1 | 4 | SQK-RAD004[a] | 7,573,765,689 | 1,216,984 | 12,320 | 11.3 |
| 2 | 5 | SQK-LSK109 | 4,282,119,674 | 392,256 | 35,584 | 11.38 |
| 2 | 6 | SQK-LSK109 | 4,889,116,279 | 538,051 | 26,881 | 12.07 |
| 2 | 7 | SQK-RAD004 | 6,913,193,761 | 1,128,659 | 18,562 | 10.58 |
| 2 | 8 | SQK-RAD004 | 8,493,017,228 | 1,830,809 | 11,868 | 10.51 |

SQK-LSK109 is the ligation based library preparation kit. SQK-RAD004 is the transposon based rapid library preparation kit.
[a]Size selection on extracted DNA, before library preparation using the Circulomics short read eliminator kit.

The miniasm parameters used for the down-selected assembly that was subsequently polished and used for genome scaffolding (Table 6, v0.0) were "-m 100 -i 0.05 -s 500 -I 0.8 -e 3." These settings are only slightly less stringent than the default settings (-m 100 -i 0.05 -s 1000 -I 0.8 -e 4), with mappings less than 500 instead of 1,000 total bases dropped (-s), and contigs generated from less than three instead of four reads removed (-e). The three parameters that remained at the default value are all more stringent compared with other parameter set values tested. The assembly was selected based on its relatively low contig count compared with that produced from other parameters sets, and a total assembly length approaching that of CanFam3.1.

Subsequent polishing of the v0.0 assembly using Racon resulted in large increases in "BUSCO complete" percentages, starting at only 0.20% in v0.0 (unpolished assembly), 32.00% in v0.1 (3× ONT polishing), and 94.80% in v0.2 (2× Illumina polishing). After contig-level scaffolding of 10X contigs of each haplotype onto v0.2, then chromosome-level scaffolding of each v0.3 haplotype onto the v0.4 scaffold, BUSCO complete percentages were further increased to 95.00% and 95.10% for v1.0a and v1.0b, respectively (Table 6). These values are comparable with those achieved by CanFam3.1 at 95.20%. Compared with the 10X SuperNova pseudohap assembly the N per 100 kb metric was much improved through scaffolding onto the polished ONT scaffold (v0.2), from 1,901 down to only 275.90 and 275.77 in the final assembly haplotypes v1.0a and b, respectively. This suggests that the contiguous regions of the final assembly haplotypes are similar, the only differences being SNPs and small indels. It is important to note that all chromosomes in either haplotype (a) or (b) were not determined to be from a single parent (maternal or paternal). In addition, the CanFam3.1 reference contains 429 N per 100 kb, significantly more than the v1.0 assembly. Although the German Shepherd assembly (GCA_008641245.1) contains only 236 N per 100 kb, it only contains 93.7% complete BUSCOs. Overall, the total length of v1.0a and v1.0b are similar, at ~2.39 Gb, with the largest contig about 10% larger than that of either CanFam3.1 or the German Shepherd assembly.

## Mapping available public sequence data against reference genomes

To evaluate performance as a new reference genome, publicly available Illumina WGS reads from 10 Labrador Retrievers were obtained from NCBI's Sequence Read Archive (SRA). These are part of a 722 canid dataset, each sequenced with Illumina WGS and deposited on SRA in 2018 (accessions available in Table S2). It is one of the first data sets to be available for researchers to explore genomic variability among canid species beyond SNP-chip-level variation (Plassais et al, 2019). Ten Labrador Retriever data sets were mapped against three different canid breed reference genomes: Boxer (CF, CanFam3.1, GCF_000002285.3), German Shepard (GS, GCA_008641245.1), and the Labrador Retriever genome presented here (YA, Yella_v1.0a, CP050567-CP050606). Fig 5 shows alignment rates and total high-quality variants called for each. In comparison with the Boxer and German Shepherd reference genomes, significantly more reads map to our Labrador Retriever reference, as expected (Fig 5A, paired $t$ test; CF versus YA $P$-value = $2.457 \times 10^{-6}$, GS versus YA $P$-value = $1.397 \times 10^{-3}$). One area in which a breed-specific reference would be expected to excel is when calling variants. Assuming that a genome specific to a breed has the most conserved structural and SNP variation, the number of called variants should decrease when reads from the same breed are mapped versus reads derived from a different breed. This can clearly be seen in Fig 5B, which shows the number of high-quality variants called (those with Q-score ≥ 30) from the 10 Labradors mapped against each reference. Interestingly, the Boxer and Shepherd show similar performance when compared with total variants called in the Labrador, with the Labrador resolving an average of ~15% of variants called against the non-Labrador breeds (Table S2 Supplemental Data 1).

## Mitochondrial sequence and Y-chromosome

The mitochondrial (MT) genome was easily recoverable from Yella and comparable with the CanFam3.1 MT reference (Fig S3). It was annotated and visualized using GeSeq (Tillich et al, 2017). The Y-chromosome was much more recalcitrant. Yella is a male Labrador Retriever, and whereas reads from the Y-chromosome could be detected via alignment to an existing partial Y chromosome reference sequence, the Y-chromosome for Yella was not able to be resolved beyond an acceptable threshold for a published reference genome. This is similar to issues experienced across mammalian genomics, in which the short and highly repetitive nature of the Y-chromosome, along with its homology to the X-chromosome can

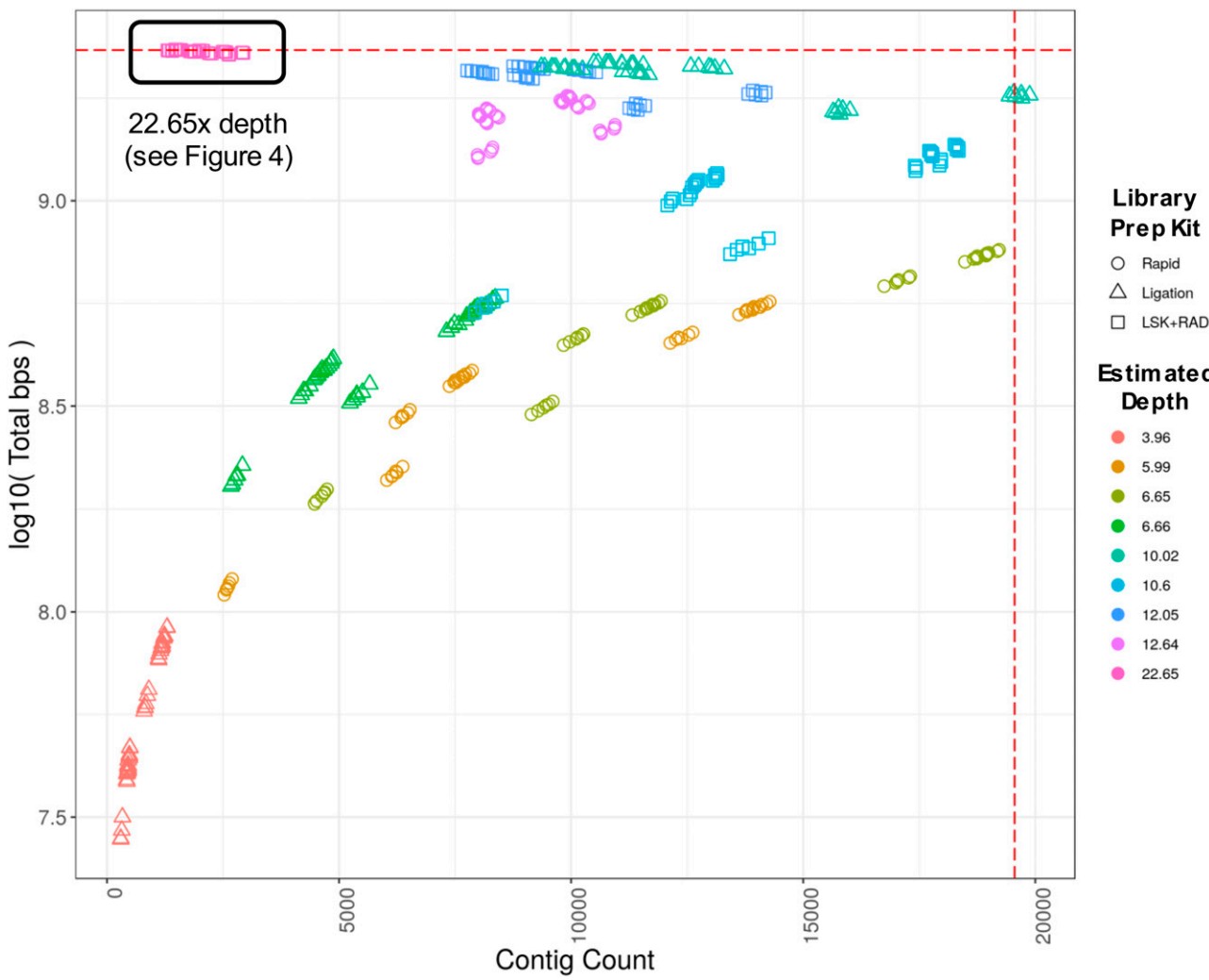

**Figure 3. Genome assembly contig count versus total length of assembly.**
Each point represents a distinct assembly resulting from one of 144 unique miniasm parameter combinations. Sequence data from eight ONT flow cells are represented in the plot, four from each of the Ligation and Rapid library preparation kits (SQK-LSK109 and SQK-RAD004, respectively). See Table 3 for details linking "estimated depth" to sequencing run and library kit. The estimated depth of 22.65 is a combination of reads from all eight flow cells (black boxed region in upper left, see Fig 4 for details regarding parameters). Estimated coverage is based on the total bps in the read set divided by the total length of CanFam3.1 assembly including Ns. Total bps of assembly approaches estimated total genome size as depth approaches 20×. Horizontal dashed red line—size of CanFam3.1 with N's (2,327,604,993 bp); vertical dashed red line—contig count (19,555) of CanFam3.1 chromosomal scaffolds broken at every occurrence of N. The following "Estimated Depth(s)" are from: the rapid kit only (5.99, 6.65, and 12.64); the ligation kit only (3.96, 6.66, 10.02); and a combination of the two (10.60, 12.05, and 22.65).

make it difficult to detect and assemble (Carvalho & Clark, 2013; Li et al, 2013; Oetjens et al, 2018; Rangavittal et al, 2019).

## Discussion

Over the past two decades, much of the population-wide haplotyping of humans and dogs necessitated using SNPs derived from a single reference genome. In both cases, the starting references (from a combined sample of European Americans and an individual Boxer, respectively) would not be useful for ethnic stratification (for humans) or breed stratification (for canids). This can lead to an influx of false positives and false negatives when calling variants for a mixed population. In addition, the reliance on SNPs has failed to capture structural variation among populations, which has also not been well captured by array methodologies. One way to address both of these issues is the generation of a

"stratified reference" with cheaper technologies, such as short-read WGS, before initiating a GWAS. Here, we provide the wet laboratory and bioinformatic methodology to generate a high-resolution mammalian reference genome for ~$10,000 (not including costs of labor for data analysis or laboratory infrastructure). Offsetting these costs would be the improved resolution of individuals mapped to the reference, and the elimination of a large proportion of variant call noise. We show that publicly available canid data generated with short-read WGS can be remapped, allowing more comparative controls to be used for a GWAS without further expenditure. Investigators using this approach could affordably generate a high-quality GWAS using a high-resolution, stratified reference, and a population genotyped using WGS. In canids, this could allow for breed-specific elucidation of structural variants, and, more importantly, the determination of their frequencies within that breed. As frequencies of SNPs and structural variants are combined,

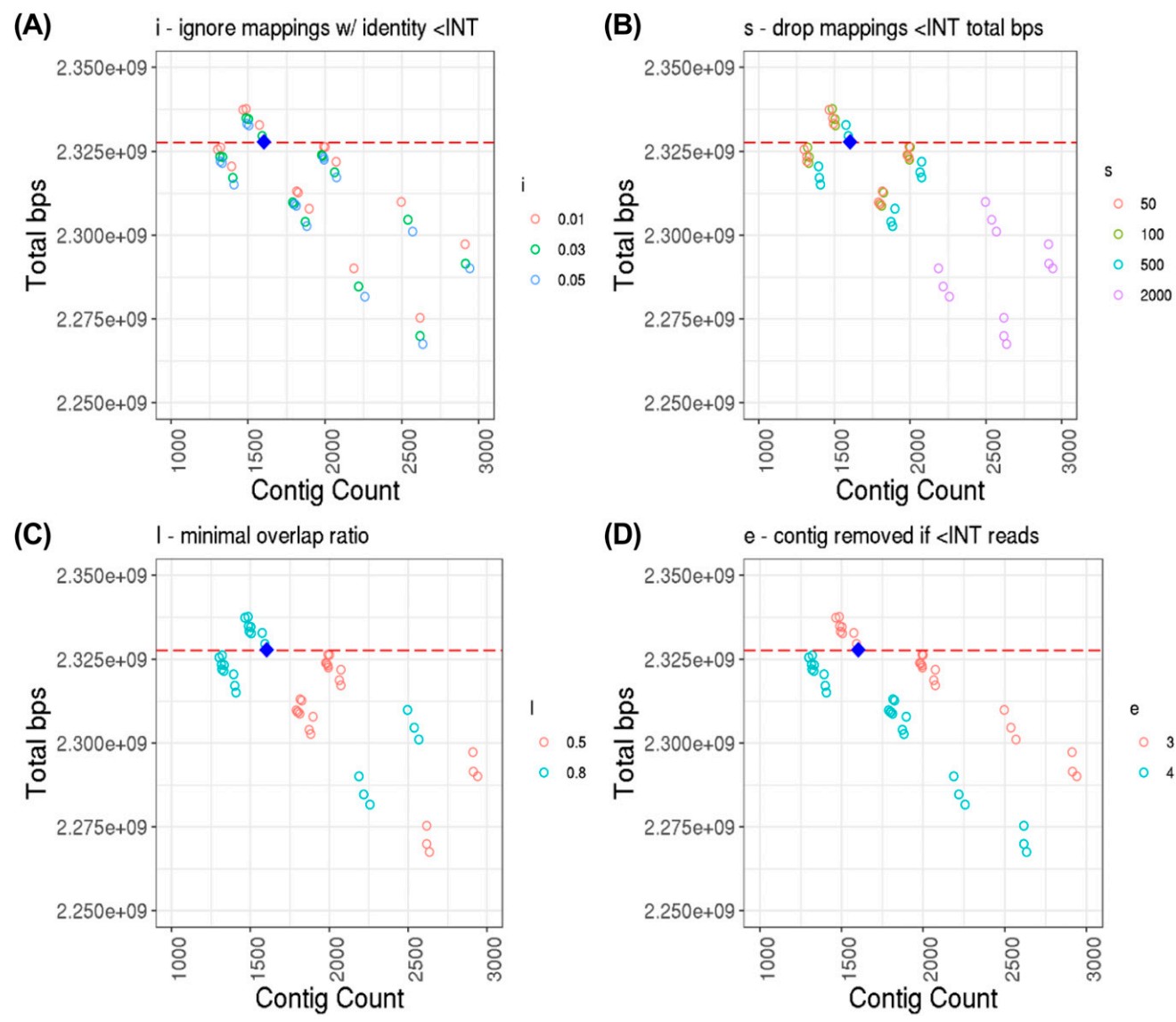

**Figure 4. Genome assembly contig count versus total length of assembly.**
The estimated genome depth of the data is 22.65×. Contig count calculated from counting number of headers in resulting assembly FASTA files, and total length calculated from non-header character count. **(A, B, C, D)** Zoomed in view of the top-left group of assemblies from Fig 3, colored by parameter value and broken down by miniasm parameter type: (A) i, ignore mappings with identity less than INT identity; (B) s, drop mapping less than INT total bps; (C) I, minimap overlap ratio; and (D) e, contig is removed if it is generated from less than INT reads. Note that miniasm parameter "m" (for dropping read mappings with less than INT matching bps) is left out, as all points for the three values used (25, 50, and 100) are all overlapping (i.e., "m" has no effect on contig count or total bps). Default parameters for miniasm are: m = 100, i = 0.05, s = 1,000, I = 0.8, e = 4. The blue diamond indicates the down-selected assembly (v0.0 in Table 4) used for polishing and final scaffolding, miniasm parameters used: m = 100, i = 0.05, s = 500, I = 0.8, e = 3. The red dashed line indicates the genome size (with N's) of CanFam3.1.

these data could then be applied towards the ultimate genomic reference goal: the *Canis lupus familiaris* pan-genome.

# Materials and Methods

### Sample collection

Blood samples were obtained from four canines, and collected in both PAXGene Blood DNA tubes (761115; PreAnalytix) and "purple top" EDTA Vacutainer tubes (367863; BD Biosciences). Blood samples were stored at 4°C upon arrival and processed within 2 d. Samples were split between

four different DNA extraction protocols (described below) to test extraction efficiency. Note that blood from only a single individual was used for genome sequencing and assembly, this was a 2-yr and 4-mo-old male Labrador Retriever.

### DNA extraction and analysis of HMW-DNA

Four DNA extraction protocols were used to process blood samples: (1) the Dog Genome Project Protocol ("Online Research Resources Developed at NHGRI," n.d.) which uses a PCE, (2) the PAXgene Blood DNA kit (761133; PreAnalytix), (3) the MagMax Core NA kit (A32700; Applied BioScience), and (4) the Nanobind CBB Big DNA Kit (Beta Ultra-High

**Table 5. Illumina 10X library, NovaSeq S1 flowcell 300 cycle sequencing run summaries.**

| Run | Lane | Paired read | RAW | | TRIMMED | |
|---|---|---|---|---|---|---|
| | | | Total bps | Total reads | Total bps | Total reads |
| 3 | 1 | 1 | 2.36E+10 | 156,607,429 | 2.00E+10 | 155,880,038 |
| 3 | 1 | 2 | 2.36E+10 | 156,607,429 | 2.35E+10 | 155,880,038 |
| 3 | 2 | 1 | 2.29E+10 | 151,709,875 | 1.94E+10 | 151,035,675 |
| 3 | 2 | 2 | 2.29E+10 | 151,709,875 | 2.27E+10 | 151,035,675 |
| 4 | 1 | 1 | 3.16E+10 | 209,187,620 | 2.68E+10 | 208,419,758 |
| 4 | 1 | 2 | 3.16E+10 | 209,187,620 | 3.14E+10 | 208,419,758 |
| 4 | 2 | 1 | 3.24E+10 | 214,451,964 | 2.75E+10 | 213,618,769 |
| 4 | 2 | 2 | 3.24E+10 | 214,451,964 | 3.22E+10 | 213,618,769 |
| Totals | | | 2.21E+11 | 1,463,913,776 | 2.03E+11 | 1,457,908,480 |
| Est. depth | | | 95.38 | | 87.80 | |

Insert size ~400 bp, these libraries were not prepared with the intention of joining (hence, the 100-bp gap between pairs). Quality and adapter trimming was performed with cutadapt (including clipping the first 22 bases from R1).

Molecular Weight DNA Extraction Protocol V1.4, Circulomics). Blood samples were split based on input requirements for each kit and processed according to the manufacturer's protocol. Nucleic acid extracts were then quantified by Qubit 4.0 using the Broad Range dsDNA kit (Q32853; Thermo Fisher Scientific), and for nucleic acid purity using the NanoDrop 2000 (Thermo Fisher Scientific). HMW-DNA (HMW DNA) was visualized using Pulsed Field Gel Electrophoresis on a Blue Pippen Pulse, set on 70 V for 20 h at room temperature. Samples were stored at –20°C until quantified for sequencing library preparation.

### ONT library preparation and sequencing

DNA from the Nanobind CBB Big DNA kit and the MagMax Core NA kit for both PAXgene and "purple top" EDTA tubes were combined to create a single sample for Oxford Nanopore Technologies library preparation. Half of this sample was used in the Short Read Eliminator Kit (SS-100-101-01; Circulomics, Inc.) to test the effect of size-selection on read N50, resulting in a size-selected sample. The size-selected and non-size-selected samples were then split between the Rapid Sequencing Kit (SQK-RAD004; Oxford Nanopore Technologies) and the Ligation Sequencing Kit (SQK-LSK109; Oxford Nanopore Technologies) to test the effect of library preparation on read N50, resulting in a total of four unique libraries. Each library was then loaded onto an R9.4.1 flow cell and sequenced in parallel on the ONT GridION platform. It was determined that size-selection did not have the desired effect of increasing read N50, and four additional non size-selected libraries were prepared (two SQK-RAD004 and two SQK-LSK109) to achieve a target depth of at least 20×. The

**Table 6. Assembly metrics of Yella dog genome through the scaffolding process, with related dog genome assembly metrics for comparison.**

| Description | Total contigs | Largest contig | Total length (Gb) | GC content | N50 (Mb) | L50 | N per 100 Kb | BUSCO scores | |
|---|---|---|---|---|---|---|---|---|---|
| | | | | | | | | Complete | Fragmented |
| CF, GCF_000002285.3 | 82 | 123,773,608 | 2.328 | 41.06% | 47.7 | 19 | 429 | 95.20% | 2.50% |
| GS, GCA_008641245.1 | 40 | 126,700,074 | 2.367 | 41.21% | 64.5 | 14 | 236 | 93.70% | 3.40% |
| CFGS, RaGOO of CF onto GS | 40 | 123,868,242 | 2.328 | 41.06% | 64.2 | 14 | 430 | 92.90% | 3.80% |
| JHMI 10X pseudohap | 10,391 | 96,528,903 | 2.417 | 41.25% | 39.2 | 22 | 1,901 | 92.70% | 4.40% |
| v0.0 | 1,601 | 20,780,228 | 2.299 | 41.11% | 5.5 | 130 | 0 | 0.20% | 1.10% |
| v0.1 | 1,600 | 21,039,211 | 2.326 | 40.98% | 5.6 | 130 | 0 | 32.00% | 21.80% |
| v0.2 | 1,600 | 21,018,819 | 2.324 | 41.17% | 5.6 | 130 | 0 | 94.80% | 2.70% |
| v0.3a | 1,412 | 21,088,418 | 2.394 | 41.30% | 5.4 | 134 | 270 | 95.20% | 2.60% |
| v0.3b | 1,413 | 21,084,388 | 2.394 | 41.30% | 5.4 | 134 | 270 | 95.20% | 2.50% |
| v0.4 | 40 | 131,668,473 | 2.435 | 41.30% | 64.9 | 14 | 1,972 | 92.40% | 4.20% |
| v1.0a | 40 | 138,659,542 | 2.394 | 41.30% | 64.3 | 14 | 276 | 95.00% | 2.50% |
| v1.0b | 40 | 138,666,786 | 2.493 | 41.30% | 64.3 | 14 | 276 | 95.10% | 2.30% |

The (a) and (b) suffixes represent the different haplotype genomes. BUSCO scores calculated using v3 with the mammalia_odb9 dataset (missing % equals 100 – [Complete + Fragmented]).

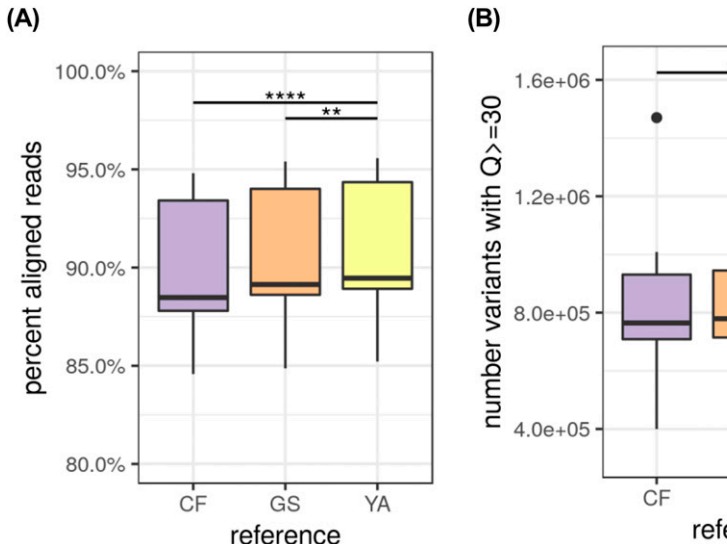

**Figure 5. Alignment rates and total variants of 10 Labrador Retriever Illumina data sets from Sequence Read Archive.**
Accessions and additional metrics can be found in Table S1. **(A)** Reads alignment rates to CF (GCF_000002285.3, CamFam3.1, Boxer breed), GS (GCA_008641245.1, German Shepherd breed), and YA (Yella v1.0; Labrador Retriever breed) reference genomes (paired $t$ test; CF versus YA $P$-value = 2.457 × $10^{-6}$, GS versus YA $P$-value = 1.397 × $10^{-3}$). **(B)** Total variants detected at Q > 29 in references (paired $t$ test; CF versus YA $P$-value = 4.744 × $10^{-6}$, GS versus YA $P$-value = 3.931 × $10^{-6}$).

output of all eight flow cells produced a combined total of ~22.7× depth.

### 10X Genomics linked-read sequencing and assembly

For the 10X Genomics assembly, HMW genomic DNA was isolated from whole blood stored in the PAXgene proprietary media using the Nanobind CBB Big DNA kit (Circulomics, Inc.) and short fragments filtered out using the Circulomics Short Read Eliminator kit. Genomic DNA concentration and purity were assessed with a Qubit 2.0 Fluorometer (Thermo Fisher Scientific) and NanoDrop 2000 spectrophotometer (Thermo Fisher Scientific). Capillary electrophoresis was carried out using a Fragment Analyzer (Agilent Technologies) to ensure that the isolated DNA had a minimum molecule length of 40 kb. Genomic DNA was diluted to ~1.2 ng/$\mu$l and libraries were prepared using Chromium Genome Reagents Kits Version 2 and the 10X Genomics Chromium Controller instrument fitted with a micro-fluidic Genome Chip (10X Genomics). DNA molecules were captured in Gel Bead-In-Emulsions (GEMs) and nick-translated using bead-specific unique molecular identifiers (Chromium Genome Reagents Kit Version 2 User Guide) and size and concentration determined using an Agilent 2100 Bioanalyzer DNA 1000 chip (Agilent Technologies). Libraries were then sequenced on an Illumina NovaSeq 6000 System using an S1 flowcell, following the manufacturer's protocols (Illumina) to produce >95× read depth using paired-end 150 bp reads. The reads were assembled into phased pseudo-haplotypes using Supernova Version 2.0 (10X Genomics).

### Genome assembly

As discussed above, two sequencing platforms were used to sequence and assemble the yellow Labrador Retriever mixed breed *Canis lupus familiaris* phased reference genome; HMW sequencing using R9.4.1 flow cells on ONT's GridION platform, and 10X Genomics linked-read sequencing on Illumina's NovaSeq

platform. The de novo assembly workflow (Fig 6) starts with generating an overlapping read file from all ONT data using minimap2 (version 2.15-r911-dirty) (Li, 2018). These super-contiguous sequences and the original input read file were then assembled using miniasm (version 0.3-r179) (Li, 2018). To find the best initial assembly for polishing and scaffolding, a range of miniasm parameter combinations were executed as part of this step, and each resulting assembly evaluated for total contig count and length. A five feature parameter space for miniasm was explored, yielding 144 unique parameter tests (see Fig 4 for specific values used for parameters m [3x], i[3x], s[4x], I[2x], and e[2x]).

After assembly down-selection (v0.0, see the Results section for specific parameter set), the raw contig correction by rapid assembly methods tool Racon (version v1.4.3) was used for polishing; three rounds with ONT reads (v0.1) followed by two rounds with Illumina 10X reads (v0.2) (Vaser et al, 2017). The read QC tool cutadapt (version 2.5) was used to clip the first 22 bps containing the GEM barcode from the Illumina 10X reads before use as polishing input (Martin, 2011). In addition, a base call quality threshold of Phred 20 and a minimum length of 50 bp were used during cutadapt QC processing. To produce phased haplotypes, the SuperNova pseudohap2.1 and 2.2 contig sets were scaffolded separately onto v0.2, producing v0.3a and b, respectively (Table 6). The fast and accurate reference-guided scaffolding tool RaGOO (version v1.1) was used to accomplish all scaffolding (Alonge et al, 2019). Alongside polishing and pseudohap phasing of the ONT scaffolds, CanFam3.1 (GCF_000002285.3) was scaffolded onto the newly assembled German Shepherd genome (GCA_008641245.1) (Field et al, 2020) because the latter provides superior chromosomal context for the more fragmented but highly annotated CanFam3.1 genome (CFGS). Next, the unphased SuperNova pseudohap1 contigs were scaffolded onto the CFGS assembly to correct for potential structural variation between breeds, and more accurately reflect the structure of the Labrador Retriever breed (v0.4). Last, a final phased v1.0a and b assembly was produced by scaffolding v0.3a and b onto v0.4. Assembly statistics were calculated using QUAST-LG (version v5.0.2),

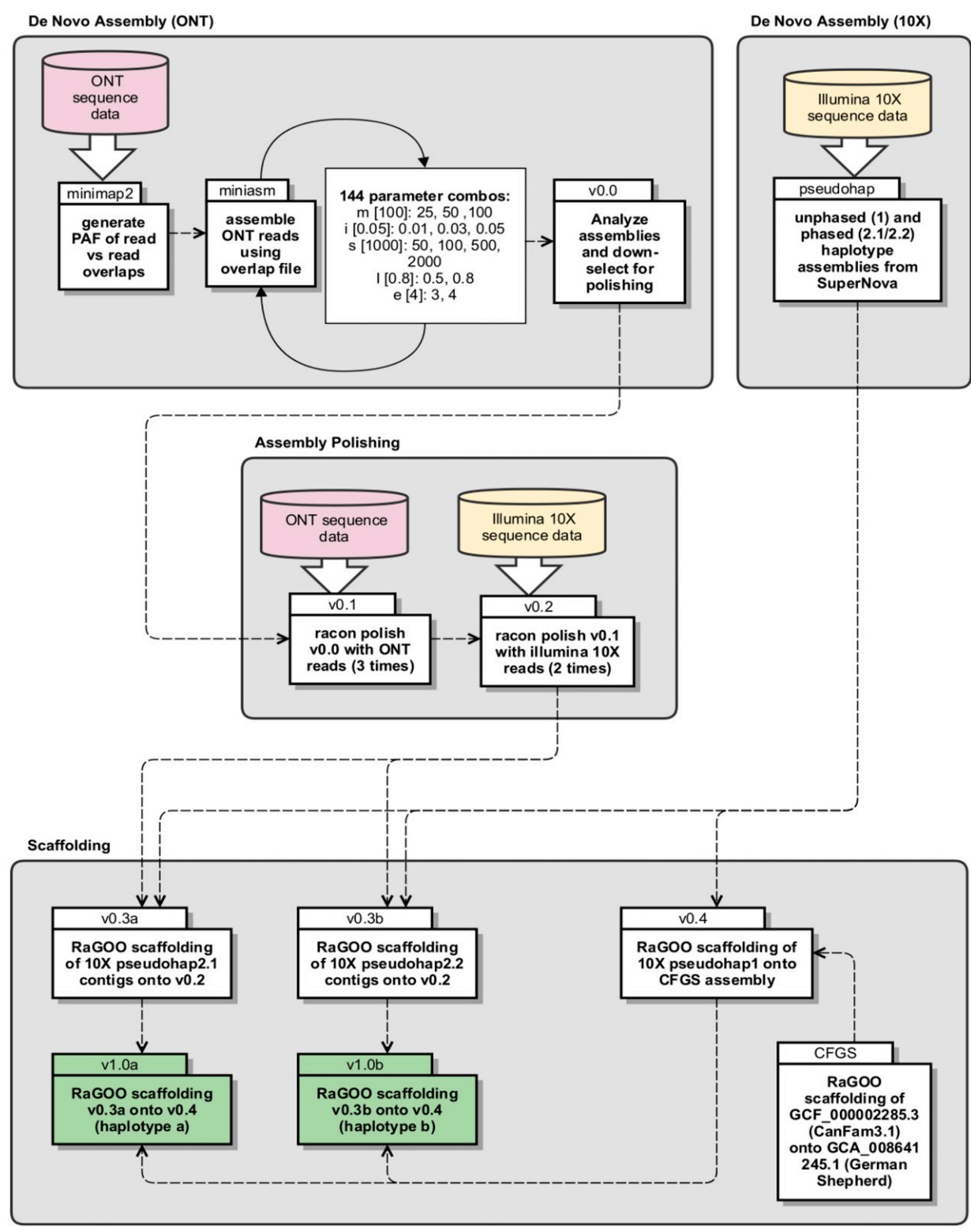

**Figure 6. Diagram of phased assembly pipeline.**
Divided into four primary sections: De Novo Assembly (Oxford Nanopore Technologies), De Novo Assembly (10X), Assembly Polishing, and Scaffolding.

and genome completeness was assessed using BUSCO (version v3, Benchmarking sets of Universal Single-Copy Orthologs) with the mammalia_odb9 dataset (https://busco.ezlab.org/datasets/mammalia_odb9.tar.gz) (Simão et al, 2015; Mikheenko et al, 2018).

### Alignment and variant calling

Reads from SRA were aligned to the three canine reference genomes shown in Fig 5 using default parameter settings for the graph-based aligner HISAT2 (Kim et al, 2019). Secondary and supplementary alignments were then filtered using samtools with parameters "-F0x4 -F0x100 -F0x800" (Li et al, 2009). Variant calling was performed using default parameters for "bcftools mpileup" and "bcftools call," then filtering out variant calls with QUAL less than 30 (Li, 2011).

## Data Availability

The sequence read data and assemblies generated in this study have been submitted to the NCBI BioProject database (https://www.ncbi.nlm.nih.gov/bioproject/) under accession number PRJNA610592. All samples used in this study are under BioSample SAMN14279123. The primary haplotype FASTAs are under BioProject PRJNA610232 and differentiated from the alternative haplotype with an "a" at the end of header names excluding the MT header (40 sequences, MT included, GenBank accessions CP050567.1 - CP050606.1). The alternative haplotype FASTAs are under BioProject PRJNA610230 and differentiated from the primary with a "b" at the end of header names (39 sequences, MT ommited, GenBank accessions CP050607.1 - CP050645.1).

## Ethics Statement

All canine blood samples were ethically collected under the Johns Hopkins Animal Care and Use Committee–approved Standard Operating Procedure #SP18P123 by a licensed veterinary doctor.

## Supplementary Information

## Acknowledgements

Karen L Meidenbauer, (DVM) Doctor of Veterinary Medicine, is acknowledged for her technical leadership and expertise, as well as her participation in drawing blood, which was transported via shippable pelican case with all lab equipment, reagents, and samples. David M Deglau and Michael A House are gratefully acknowledged for project and program management, respectively. Jody BG Proescher is acknowledged for her critical review of and editorial feedback for the manuscript. Funding for this project was provided by the Department of Homeland Security Science and Technology (S&T) Directorate, Contract No. 70RSAT19CB0000002.

### Author Contributions

RA Player: data curation, software, formal analysis, investigation, methodology, and writing—original draft, review, and editing.
ER Forsyth: laboratory methodology.
KJ Verratti: laboratory methodology.
DW Mohr: data curation, formal analysis, and methodology.
AF Scott: data curation, formal analysis, funding acquisition, and methodology.
CE Bradburne: conceptualization, formal analysis, supervision, funding acquisition, investigation, methodology, project administration, and writing—original draft, review, and editing.

### Conflict of Interest Statement

The authors declare no conflict of interest. Distribution statement A: Approved for public release; distribution is unlimited.

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
