## [Reviewer comments · Life Science Alliance]

Life Science Alliance

A Novel *Canis lupus familiaris* Reference Genome Improves Variant Resolution for Breed-Specific GWAS.

Robert Player, Ellen Forsyth, Kathleen Verratti, David Mohr, Alan Scott, and Christopher Bradburne DOI: <https://doi.org/10.26508/lsa.202000902>

Corresponding author(s): Christopher Bradburne, The Johns Hopkins University Applied Physics Laboratory

Review Timeline:

Submission Date:	2020-09-03
Editorial Decision:	2020-12-17
Revision Received:	2020-12-28
Editorial Decision:	2021-01-04
Revision Received:	2021-01-12
Accepted:	2021-01-13

Scientific Editor: Shachi Bhatt

Transaction Report:

December 17, 2020

Re: Life Science Alliance manuscript #LSA-2020-00902-T

Christopher E. Bradburne
The Johns Hopkins University Applied Physics Laboratory

Dear Dr. Bradburne,

Thank you for submitting your manuscript entitled "A Phased Canis lupus familiaris Labrador Retriever Reference Genome Utilizing High Molecular Weight DNA Extraction Methods and High Resolution Sequencing Technologies" to Life Science Alliance. The manuscript was assessed by expert reviewers, whose comments are appended to this letter.

As you will see from the reviewers' comments, both reviewers are quite enthusiastic about the findings, and have recommended only minor edits for revisions. We thus encourage you to submit a revised version back to us that addresses the reviewers' points.

Please accept my sincerest apologies for the delay in the handling of this manuscript.

Thank you for this interesting contribution to Life Science Alliance. We are looking forward to receiving your revised manuscript.

Sincerely,

Shachi Bhatt, Ph.D.
Executive Editor

- A letter addressing the reviewers' comments point by point.
- An editable version of the final text (.DOC or .DOCX) is needed for copyediting (no PDFs).
- High-resolution figure, supplementary figure and video files uploaded as individual files: See our detailed guidelines for preparing your production-ready images, <https://www.life-science-alliance.org/authors>
- Summary blurb (enter in submission system): A short text summarizing in a single sentence the study (max. 200 characters including spaces). This text is used in conjunction with the titles of papers, hence should be informative and complementary to the title and running title. It should describe the context and significance of the findings for a general readership; it should be written in the present tense and refer to the work in the third person. Author names should not be mentioned.

B. MANUSCRIPT ORGANIZATION AND FORMATTING:

Reviewer #1 (Comments to the Authors (Required)):

Player et al. constructed a phased Labrador Retriever genome assembly based on Oxford Nanopore (ONT) and 10xGenomics (10X) sequencing data. The authors provide a limited but useful evaluation of different wet lab and bioinformatic parameters to optimize the quality of the resulting genome assembly. The genome assembly of an individual dog (4 or 5 others are also publicly available) is of high value to the dog scientific community. Furthermore, the methodological approach is innovative and the properties of this approach are adequately discussed. The authors convincingly demonstrate that their approach enables the generation of a de novo assembly of good quality with a relatively small budget that is in reach to most academic research groups. The presentation of the data should be improved in a few instances.

Specific comments:

(1)

Title: What are "high resolution" sequencing technologies? The resolution is at the single base level for all sequencing technologies. Consider to replace with long-read sequencing technologies or explicitly state that you used a combination of ONT and 10x.

(2)

lines 32/33: The costs for the new assembly are only mentioned in the abstract, but not in the rest of the manuscript. This cost should be either deleted or explained in a little bit more detail in the discussion of the manuscript. I assume that the stated 10,000 USD represent the costs for the consumables, but do not include the costs for work including data analysis or laboratory infrastructure.

(3)

Line 56 and throughout the manuscript. The citations of website references look very awkward ("n.d.").

(4)

Lines 66-68: The CanFam3.1 assembly is mostly based on Sanger shotgun sequencing with limited illumina polishing. The CanFam3.1 assembly has not changed since its release in 2011. (The annotation improved, but the assembly did not change.) There are now newer and better canine assemblies available (e.g. UU_Cfam_GSD_1.0/canFam4), but these newer assemblies have not yet replaced the widely used CanFam3.1 as the standard genome reference.

(5)

Line 70: I am not aware of any SNP chip that costs USD 500 per dog. I suggest to revise into "approximately USD 100" per animal or possibly USD 100-200, if you want to be generous.

(6)

Line 88: Please state here that the donor for the genome was male.

(7)

Line 90: I suggest to delete "or breeders".

(8)

Line 92: What is the meaning of "local"? Consider to delete.

(9)

Table 1: The total recovered DNA amount should be normalized to the input volume of blood (e.g. DNA yield per 1 ml of blood).

(10)

Table 2: The total NA should be given per ml of input blood.

(11)

It is "surprising" to see an NA quality (260/280) of 5.21 for the PCE method applied to Yella's sample in table 1 compared to a mean of 1.75 for all four isolated DNA samples. This suggests a gross error during the processing of Yella's sample. This striking discrepancy needs more

explanation/discussion (either in the methods or in the supplementary data).

(12)

The method to determine the presence of high molecular weight DNA (yes/no) is not clear to me. I am not aware of a BluePippin Pulse instrument. I am aware of a Fragment Analyzer Pulse instrument that would be perfectly suitable to accurately determine the size distribution of the isolated DNA. There also is a BluePippin instrument used for preparative size selection of DNA fragments. I am not sure whether this instrument would allow the required analytical assessment of the fragment sizes. Finally, the supplementary figure S1 looks like a conventional pulsed-field agarose gel to me. If so, then the used method must be faithfully described (including pulse angle and pulse duration).

(13)

Line 139: State the chemistry (2 x 150 bp ?) and the type of flow cell used. Four lanes on which type of flow cell? In the abstract it is stated that one lane of an S1 flow cell would be sufficient. However, an S1 flow cell has only 2 lanes. Double-check whether you used S1 or S4 flow cells (or possibly yet another flow cell type).

(14)

Also line 139: Before you state the specifics of the illumina sequencing run(s), you must briefly state the library preparation (10XGenomics). Library preparation comes prior to sequencing!

(15)

Table 5: I suggest to not give identical values for the forward and reverse reads. The table would be much easier to read, if you have only 4 rows of data stating the stats for the read-pairs.

(16)

Are the v1.0a and v1.0b genome assemblies the two haplotypes of Yella? If so, this should be more clearly stated, e.g. in Table 6.

(17)

Line 265: To the best of my knowledge the human genome reference is not derived from a single individual but from a pool of 8 donors. Please double-check the accuracy of your statement.

(18)

Line 273: canids -> canid genomes (or canid WGS data)

(19)

Lines 282-287. An ethics statements for the collection of blood samples is missing. More information on the 4 sampled dogs should be given (sex, breed, age).

(20)

Lines 296-299: Methodology for pulsed-field gel electrophoresis needs to be revised/expnaded (see my comment no. 12)

(21)

Line 320: Fragment Analyzer or Fragment Analyzer Pulse?

(22)

Line 326: Specify the flow cells (and possibly the splitting into individual lanes) for the NovaSeq

6000 sequencing.

(23)

Line 340: Figure x ???

(24)

Line 397-398: Did AS fund the experiments (privately) or was he responsible for funding acquisition?

(25)

Figure S3: In my version of the manuscript, the labels on the mtDNA genomes were too small and too blurred to be readable. Consider to revise the figure. What are the differences between Tasha's (=CanFam3.1) and Yella's mtDNA?

(26)

Table S1: To which Kit Input Volume is the DNA amount normalized?

(27)

How can you be sure that all the "a-chromosomes" belong to one haplotype (paternal or maternal) and all the "b-chromosomes" to the other parental haplotype? If it is not clear that all "a-chromosomes" correspond to one particular parental haplotype, then this should be clearly stated somewhere in the manuscript.

Reviewer #2 (Comments to the Authors (Required)):

This manuscript describes a wet lab method and sequencing approach for generating high quality complete reference genome. The combination of long and short read sequencing methods result in a "high resolution" approach demonstrated for *Canis lupus*, but the concept and approach is transferable to other target genomes (e.g. humans).

One aspect of the work is optimizing the DNA extraction protocol for longer read sequencing technologies. The results in Table 1 and 2 are useful to others evaluating these extraction techniques. A comment on the "total amount of DNA" recovered - was this normalized to the total volume that went into the extraction method? It might provide more context for the reader to understand the recovery normalized to volume input into the extraction process.

It might be helpful to more clearly identify what extraction methods was "best" for your purposes? From the methods it looks like Magmax and Nanobind fractions were combined (from both collection tubes). I am assuming that the 260/280 was poor for PCE and that PAXgene was low MW DNA (this is shown figures 2, but the final choice made for sequencing could be clearer in the text).

The metrics for the sequence data and improvements made to variant calls and mapping are clear (although I am not an expert in this area).

I recommend that this manuscript be accepted after addressing these minor revisions.

Please see the reviewer comments listed below, along with our responses in red. All updated manuscript, tables, and figures have been uploaded, containing the requested revisions unless otherwise noted in our responses.

Reviewer #1 (Comments to the Authors (Required)):

Player et al. constructed a phased Labrador Retriever genome assembly based on Oxford Nanopore (ONT) and 10xGenomics (10X) sequencing data. The authors provide a limited but useful evaluation of different wet lab and bioinformatic parameters to optimize the quality of the resulting genome assembly. The genome assembly of an individual dog (4 or 5 others are also publicly available) is of high value to the dog scientific community. Furthermore, the methodological approach is innovative and the properties of this approach are adequately discussed. The authors convincingly demonstrate that their approach enables the generation of a de novo assembly of good quality with a relatively small budget that is in reach to most academic research groups. The presentation of the data should be improved in a few instances.

Specific comments:

(1)

Title: What are "high resolution" sequencing technologies? The resolution is at the single base level for all sequencing technologies. Consider to replace with long-read sequencing technologies or explicitly state that you used a combination of ONT and 10x.

To address this critique, and to add additional description to the paper, we have changed the title to: " A Phased *Canis lupus familiaris* Labrador Retriever Reference Genome Generated Using Optimized High Molecular Weight DNA Extraction Methods and Long-Read Sequencing Techniques Improves Variant Resolution for Breed-Specific GWAS."

(2)

lines 32/33: The costs for the new assembly are only mentioned in the abstract, but not in the rest of the manuscript. This cost should be either deleted or explained in a little bit more detail in the discussion of the manuscript. I assume that the stated 10,000 USD represent the costs for the consumables, but do not include the costs for work including data analysis or laboratory infrastructure.

That is correct that the estimated cost does not include labor for analysis or lab infrastructure. We have included the following in parenthesis at the end of the sentence where the 10K price is referenced, in the Discussions section: "(not including costs of labor for data analysis or laboratory infrastructure)"

(3)

Line 56 and throughout the manuscript. The citations of website references look very awkward ("n.d.").

These citations in the text were automatically generated using Zotero. The "n.d." stands for "no date", and is inserted where no date may be scraped from the html of a website. If it pleases LSA, we would like to leave this to the editor to decide to keep or remove.

(4)

Lines 66-68: The CanFam3.1 assembly is mostly based on Sanger shotgun sequencing with limited illumina polishing. The CanFam3.1 assembly has not changed since its release in 2011. (The annotation

improved, but the assembly did not change.) There are now newer and better canine assemblies available (e.g. UU_Cfam_GSD_1.0/canFam4), but these newer assemblies have not yet replaced the widely used CanFam3.1 as the standard genome reference.

Thank you for the clarification, we have addressed this by changing the cited sentence to: " It was sequenced mostly based on Sanger shotgun sequencing with limited Illumina polishing, and the annotations have been continuously updated".

(5)

Line 70: I am not aware of any SNP chip that costs USD 500 per dog. I suggest to revise into "approximately USD 100" per animal or possibly USD 100-200, if you want to be generous.

We have performed (unpublished) a large market survey of genotyping methods and costs, and can confirm the price range cited in the manuscript.

(6)

Line 88: Please state here that the donor for the genome was male.

We have included the sex of the donor in the indicated sentence.

(7)

Line 90: I suggest to delete "or breeders".

We respectfully decline this suggestion, as there are breeders actively participating in such studies.

(8)

Line 92: What is the meaning of "local"? Consider to delete.

We have deleted "local" from the indicated sentence.

(9)

Table 1: The total recovered DNA amount should be normalized to the input volume of blood (e.g. DNA yield per 1 ml of blood).

We have included this as an additional column in Table 1 titled "NA Total per mL Blood".

(10)

Table 2: The total NA should be given per ml of input blood.

Table 2 is focused on the variability of extracted total NA in the extraction methods, therefore we do not think it necessary to transform these data to different units.

(11)

It is "surprising" to see an NA quality (260/280) of 5.21 for the PCE method applied to Yella's sample in table 1 compared to a mean of 1.75 for all four isolated DNA samples. This suggests a gross error during the processing of Yella's sample. This striking discrepancy needs more explanation/discussion (either in the methods or in the supplementary data).

We thank the reviewer for this excellent point. Indeed, the 5.21 ratio is much higher than should normally be expected. We believe this is likely due to residual phenol that was extracted along with the aqueous

phase. Residual phenol is known to overestimate the amount of DNA in a sample, and this would explain the 5.21 reading. Generally, the PCE 260/280 readings for samples were mostly around 2, but with 3 of the samples they were outside of the normal range. Since the PCE method is not as standardized as the column or kit based methods, we expect to see this variation, and it further supports the use of a more standardized kit versus the PCE method. Additional text has been added to the Table 1 legend to explain this discrepancy.

(12)

The method to determine the presence of high molecular weight DNA (yes/no) is not clear to me. I am not aware of a BluePippin Pulse instrument. I am aware of a Fragment Analyzer Pulse instrument that would be perfectly suitable to accurately determine the size distribution of the isolated DNA. There also is a BluePippin instrument used for preparative size selection of DNA fragments. I am not sure whether this instrument would allow the required analytical assessment of the fragment sizes. Finally, the supplementary figure S1 looks like a conventional pulsed-field agarose gel to me. If so, then the used method must be faithfully described (including pulse angle and pulse duration).

We have added additional details, and the run parameters in the S1 figure legend about the instrument. The instrument we used was the Pippin Pulse (Sage Science, PPI- 0200), which is a Pulsed-Field Power Box. For the gels in S1, the run parameters were 70V for 20 hours at 4C. The gel boxes used with the Pippin Pulse do not use pulse angles, it pulses back and forth rather than side to side. Below was the program used, 70V with an initial cycle of 300msec forward pulse followed by a 100msec reverse pulse. At each step 30msec was added to the forward pulse and 10msec was added to the reverse pulse for a total of 45 steps per cycle. The cycles repeat for the time-frame given which was 20 hours.

V = 70 – waveform amplitude

A = 300 – forward time at start of run (msec)

B = 100 – reverse time at start of run (msec)

C = 30 – increment added to A at each step (msec)

D = 10 – increment added to B at each step (msec)

E = 30 – increment added to C at each step (msec)

F = 10 – increment added to D at each step (msec)

G = 45 – number of steps per cycle

(13)

Line 139: State the chemistry (2 x 150 bp ?) and the type of flow cell used. Four lanes on which type of flow cell? In the abstract it is stated that one lane of an S1 flow cell would be sufficient. However, an S1 flow cell has only 2 lanes. Double-check whether you used S1 or S4 flow cells (or possibly yet another flow cell type).

The missing information has been added, and the sentence has been changed to: "The estimated average genome depth for trimmed 2x150 bp reads from the prepared 10X Genomics library run on an Illumina NovaSeq 6000 S1 flowcell was 87.80x (Table 5)."

(14)

Also line 139: Before you state the specifics of the illumina sequencing run(s), you must briefly state the library preparation (10XGenomics). Library preparation comes prior to sequencing!

See above edit.

(15)

Table 5: I suggest to not give identical values for the forward and reverse reads. The table would be much easier to read, if you have only 4 rows of data stating the stats for the read-pairs.

There are differing "Total bps" for trimmed reads between R1 and R2, which is why there are no changes.

(16)

Are the v1.0a and v1.0b genome assemblies the two haplotypes of Yella? If so, this should be more clearly stated, e.g. in Table 6.

The following has been added to the Table 6 legend: "The (a) and (b) suffixes represent the different haplotype genomes."

(17)

Line 265: To the best of my knowledge the human genome reference is not derived from a single individual but from a pool of 8 donors. Please double-check the accuracy of your statement.

Sentence changed to: "In both cases, the starting references (from a combined sample of European Americans and an individual Boxer, respectively) would not be useful for ethnic stratification (for humans) or breed stratification (for canids)."

(18)

Line 273: canids -> canid genomes (or canid WGS data)

Thank you, should be canid WGS data, has been corrected.

(19)

Lines 282-287. An ethics statements for the collection of blood samples is missing. More information on the 4 sampled dogs should be given (sex, breed, age).

Only blood from one of these was used for sequencing and subsequent assembly. Sex and breed have been given and age added in the following sentence added to the end of the "Sample collection" subsection: "Note that blood from only a single individual was used for genome sequencing and assembly, this was a two years and four months old male Labrador Retriever."

(20)

Lines 296-299: Methodology for pulsed-field gel electrophoresis needs to be revised/expnaded (see my comment no. 12)

We have added the details and instrument in the Figure S1 legend.

(21)

Line 320: Fragment Analyzer or Fragment Analyzer Pulse?
Fragment Analyzer.

(22)

Line 326: Specify the flow cells (and possibly the splitting into individual lanes) for the NovaSeq 6000 sequencing.

An S1 flowcell was used, this has been incorporated into the sentence.

(23)

Line 340: Figure x ???

Apologies, Figure 4, has been corrected.

(24)

Line 397-398: Did AS fund the experiments (privately) or was he responsible for funding acquisition?

(25)

Figure S3: In my version of the manuscript, the labels on the mtDNA genomes were too small and too blurred to be readable. Consider to revise the figure. What are the differences between Tasha's (=CanFam3.1) and Yella's mtDNA?

A higher resolution will be provided in pdf format upon submission of the revised manuscript. The differences are listed in the figure legend: "Alignment of Yella MT to refseq MT reveals 3 bps of insertions and 74 bps of deletion (alignment CIGAR: 2678M1I17233M2I6327M50D36M24D379M). Needleman-Wunsch pairwise alignment results 99.41 identity and similarity between these MT sequences."

(26)

Table S1: To which Kit Input Volume is the DNA amount normalized?

Not input volume, should be isolate volume. To be clear, the amount is normalized to the associated kit isolate (or elution) volume. For example, the PCE the normalization factor is 1000uL, for Magmax it is 90uL, etc. The column header has been changed to "Total NA Normalized to Kit Isolation Volume (ng)".

(27)

How can you be sure that all the "a-chromosomes" belong to one haplotype (paternal or maternal) and all the "b-chromosomes" to the other parental haplotype? If it is not clear that all "a-chromosomes" correspond to one particular parental haplotype, then this should be clearly stated somewhere in the manuscript.

Excellent point, the following has been added to the paragraph just after Table 6: "It is important to note that all chromosomes in either haplotype (a) or (b) were not determined to be from a single parent (maternal or paternal). "

Reviewer #2 (Comments to the Authors (Required)):

This manuscript describes a wet lab method and sequencing approach for generating high quality complete reference genome. The combination of long and short read sequencing methods result in a "high resolution" approach demonstrated for *Canis lupus*, but the concept and approach is transferable to other target genomes (e.g. humans).

One aspect of the work is optimizing the DNA extraction protocol for longer read sequencing technologies. The results in Table 1 and 2 are useful to others evaluating these extraction techniques. A comment on the "total amount of DNA" recovered - was this normalized to the total volume that went into the extraction method? It might provide more context for the reader to understand the recovery normalized to volume input into the extraction process.

Per comments from Reviewer #1, this has been added to Table 1 under column header "NA Total per mL Blood".

It might be helpful to more clearly identify what extraction methods was "best" for your purposes? From the methods it looks like Magmax and Nanobind fractions were combined (from both collection tubes). I am assuming that the 260/280 was poor for PCE and that PAXgene was low MW DNA (this is shown figures 2, but the final choice made for sequencing could be clearer in the text).

We explicitly call out the Nanobind kit as the "best", however we required more material for completing the size-selection experiments thus the combination of it with the Magmax extract. This in the last sentence of the Result subsection "Preservation, extraction, and acquisition of HMW-DNA": "Direct comparison of extraction kits showed that the Nanobind kit provided the most consistent DNA yield and quality among the four kits tested using blood stored in EDTA from four different canines (Table 2 and Figure 2).

The metrics for the sequence data and improvements made to variant calls and mapping are clear (although I am not an expert in this area).

I recommend that this manuscript be accepted after addressing these minor revisions.

Thank you!

January 4, 2021

RE: Life Science Alliance Manuscript #LSA-2020-00902-TR

Dr. Christopher E. Bradburne
The Johns Hopkins University Applied Physics Laboratory
Asymmetric Operations Sector
11100 Johns Hopkins Rd
Laurel, Maryland 21042

Dear Dr. Bradburne,

Thank you for submitting your revised manuscript entitled "A Novel *Canis lupus familiaris* Reference Genome Improves Variant Resolution for Breed-Specific GWAS.". We would be happy to publish your paper in Life Science Alliance pending final revisions necessary to meet our formatting guidelines.

I understand that you revised the manuscript in accordance to Reviewer 1's concern, however the new title seems too long and I would request you to revert back to the original one. As for Reviewer 1's concern about some of the reference format (i.e. "n.d."), I understand your explanation, and we can leave them in the manuscript as is.

Along with the points listed below, please also attend to the following,

- please add a Category for your manuscript in our system
- please upload your main and supplementary figures as single files
- please check your figure callouts in your main manuscript text: please add callouts for Figure 2 A and B to your main manuscript text
- please provide an ethics statement for collecting the blood samples
- please provide the original unedited data for Figure S1

A. FINAL FILES:

B. MANUSCRIPT ORGANIZATION AND FORMATTING:

Sincerely,

Shachi Bhatt, Ph.D.
Executive Editor
Life Science Alliance

<https://www.lsjournal.org/>
Tweet @SciBhatt @LSAJournal

January 13, 2021

RE: Life Science Alliance Manuscript #LSA-2020-00902-TRR

Dr. Christopher E. Bradburne
The Johns Hopkins University Applied Physics Laboratory
Asymmetric Operations Sector
11100 Johns Hopkins Rd
Laurel, Maryland 21042

Dear Dr. Bradburne,

Thank you for submitting your Research Article entitled "A Novel *Canis lupus familiaris* Reference Genome Improves Variant Resolution for Breed-Specific GWAS.". It is a pleasure to let you know that your manuscript is now accepted for publication in Life Science Alliance. Congratulations on this interesting work.

DISTRIBUTION OF MATERIALS:

Again, congratulations on a very nice paper. I hope you found the review process to be constructive and are pleased with how the manuscript was handled editorially. We look forward to future exciting submissions from your lab.

Sincerely,

Shachi Bhatt, Ph.D.

Executive Editor

Life Science Alliance

<https://www.lsjournal.org/>
